



# Assessing inter-annual variability in nitrogen sourcing and retention through hybrid Bayesian watershed modeling

Jonathan W. Miller[1], Kimia Karimi[2], Arumugam Sankarasubramanian[1], Daniel R. Obenour[1]

[1] Dept. of Civil, Construction and Environmental Engineering, North Carolina State University, Raleigh, NC, USA
[2] Dept. of Geospatial Analytics, North Carolina State University, Raleigh, NC, USA

*Correspondence to*: Jonathan W. Miller (jwmille7@ncsu.edu)

**Abstract.** Excessive nutrient loading is a major cause of water quality problems worldwide, including in North Carolina (NC), where reservoirs and coastal systems are often subject to excessive algae and hypoxia. Efficient nutrient management requires that loading sources are accurately quantified. However, loading rates from various urban and rural non-point sources remain
highly uncertain especially with respect to climatological variation. Furthermore, statistical calibration of loading models does not always yield plausible results, given the noisiness and paucity of observational data common to many locations. To address these issues, we leverage data for two large NC Piedmont basins collected over three decades (1982-2017) using a mechanistically parsimonious watershed loading and transport model calibrated within a Bayesian hierarchical framework. We explore temporal drivers of loading by incorporating annual changes in precipitation, land use, livestock, and point sources
within the model formulation. Also, different representations of urban development are compared based on how they constrain model uncertainties. Results show that urban lands built before 1980 are the largest source of nutrients, exporting over twice as much nitrogen per hectare than agricultural and post-1980 urban lands. In addition, pre-1980 urban lands are the most hydrologically constant source of nutrients, while agricultural lands show the most variation among high and low flow years. Finally, undeveloped lands export an order of magnitude (~ 7-13x) less nitrogen than built environments.

## 1 Introduction

Eutrophication stimulated by anthropogenic nutrient loading is a common cause of water quality problems worldwide (Smith et al., 1999).  In North Carolina (NC, USA), watershed-level nutrient management strategies have been developed for major reservoirs like Jordan Lake (JL) and Falls Lake (FL) using various process-based models (NC DWR, 2009; Tetra Tech 2014). Such models can operate on fine temporal scales (i.e., days) and characterize various mechanistic processes related to the
transfer of water and nutrients through watersheds. However, due to the large number of uncertain parameters (i.e., rates, coefficients) included in these models, multiple parameter sets may appear to fit the observational data equally well (Beven, 2006) without benefits to predictive performance (Jackson-Blake et al., 2017). Related to these issues, there is critical need for systematic model calibration and uncertainty quantification, if modeling results are to inform management decisions (Reckhow, 1994; NRC, 2001).





Hybrid watershed models, which represent nutrient loading, transport, and retention with fewer parameters but using probabilistic calibration techniques, have also been developed for nutrient source apportionment. For example, numerous applications of the SPAtially Referenced Regressions of contaminant transport On Watershed attributes (SPARROW) models have been applied throughout the US (Preston et al., 2011; Hoos and McMahon, 2009; Garcia et al., 2011). SPARROW is calibrated in a statistical framework that allows for parameter uncertainty quantification (i.e. confidence intervals). A limitation

of SPARROW is that it models long-term average conditions, and does not directly consider variability due to changes in precipitation (e.g., wet versus dry years) and watershed development, which have been shown to greatly affect nutrient loading (Howarth et al., 2012; Sinha and Michalak, 2016; Strickling and Obenour, 2018).

Methodological enhancements to SPARROW and similar hybrid watershed models have been proposed over time (Qian et al., 2005; Wellen et al., 2012; Xia et al., 2016). Recently, a Bayesian hierarchical hybrid watershed model was developed to

leverage temporal (interannual) variability in source distributions, precipitation, and nutrient loading to improve source characterization (Strickling and Obenour, 2018). By modeling interannual variability over multiple decades, this approach provides an assessment of how land use change and hydroclimatological variations have affected nutrient loading. Additionally, it systematically incorporates and updates prior information from previous studies through Bayesian inference, which helps reduce parameter and prediction uncertainty (Strickling & Obenour, 2018).

The goal of this study is to improve our understanding of nitrogen export within two highly managed NC basins that feed critical water supply reservoirs using over 30 years of loading data (1982-2017). We characterize nitrogen export from different land uses, livestock, and point sources, and test nitrogen loading rates from different types of urban lands based on their density and the age of urbanization. In addition, we determine variations in annual nitrogen loadings and retention rates based on hydro-climatological conditions. Compared to Strickling and Obenour (2018), we focus on a smaller study area with

relatively dense monitoring, improve and compare different measures of land urbanization, and more realistically quantify inter-annual loading variations due to precipitation. Finally, we characterize instream nutrient retention rates and partition loading into various upstream sources based on varying hydro-climatological conditions to help inform current management decisions.

## 2 Methods

### 2.1 Study Area

JL and FL, located in the Piedmont region of NC (Fig. 1), were impounded by the US Army Corps of Engineers in the early 1980s. Portions of each reservoir have exceeded NC water quality criteria, particularly for algae (chlorophyll a; NC DWR, 2020). JL watershed planning has been ongoing since the early 2000s, and initial TN reductions were set at 35% for the New Hope (NH) Creek basin and 8% for the Haw River (HR) basin. FL watershed planning was formalized in 2011, and Phase I

goals of the Falls Lake Rules included a TN reduction of 40% from major sources in the watershed (http://portal.ncdenr.org/web/fallslake/). JL and FL fall within the Cape Fear River basin and Neuse River basin respectively,



but both share similar underlying hydroclimatic and soil conditions and comparable levels of anthropogenic development (Markewich et al., 1990; Strickling & Obenour, 2018).

## 2.2 Load monitoring sites (LMSs)

Nutrient load monitoring sites (LMSs) were identified based on locations that had sufficient flow and nutrient sampling data to calculate yearly TN loads. To be included, a site needed a minimum of five years of daily flow records and at least 50 water quality samples during that period of record. These minimum conditions are consistent with previous studies using WRTDS load estimates and model defaults (Hirsch and De Cicco, 2015; Chanat et al., 2012). All flow data were obtained from the United States Geological Survey (USGS), whereas nutrient data were obtained from the Water Quality Portal (WQP; Read et

al., 2017) as well as local city managers (e.g., city of Durham). The two largest sources of nutrient data (from the WQP) came from the USGS and the NC Department of Environmental Quality (NCDEQ). Sites from these different entities were often located in close proximity. Data from water quality sites with less than 5% deviations in watershed area and no intervening point sources were compiled together (Table 1).

In many cases, ample water quality data were available at the location of the USGS flow monitoring station. However, if little

or no water quality data were located at the flow station, a nearby water quality station was used instead, assuming there was less than a 20% change in watershed area between the flow and water quality monitoring stations. If multiple water quality sites were located close to the flow station, only the site with the longest record was chosen. In one exception, two water quality sites utilized the same flow monitoring station (NH1 and NH6; Table 1), which was done to include two substantial data records collected above and below a major point source on Morgan Creek. In such cases, the LMSs were represented at

the location of the water quality monitoring sites, and flows were adjusted based on the drainage area ratio between the two sites adjusting for any intervening wastewater flow.

There were 25 LMSs in our study area (Fig. 1; Table 1). Stations were split into three major basins for classification purposes: the Haw River basin of JL, the NH Creek basin of JL, and the FL basin. LMSs captured 85% of the HR basin, 49% of the NH basin, and 62% of the FL basin. Three LMSs were located directly downstream of major impoundments (HR4, FL6, and FL

9; Fig. 1; Table 1).

## 2.3 Watershed Delineations

LMS watersheds were delineated using Spatial Analyst tools in ArcMap 10.6.1 (ESRI, 2018). Watershed drainage areas ranged from 11 km$^2$ to over 3000 km$^2$ (Table 1) with a median value of 106 km$^2$. Often, LMS watersheds had one or more LMS contained within their upstream watershed (Fig. 1). In order to accommodate nested LMS watersheds, we determined

incremental LMS watersheds by subtracting out any upstream LMS watersheds that were contained in a larger (downstream) LMS watershed. If a LMS did not have an upstream LMS in its watershed, its incremental watershed was equal to its total watershed.



To more accurately account for nitrogen transport and retention, incremental LMS watersheds were divided into subwatersheds (Fig. 1). Most data (e.g., land use, precipitation, livestock) were compiled at the subwatershed level. The largest possible

subwatershed corresponded to a USGS 12-digit hydraulic unit code (HUC; https://water.usgs.gov/GIS/huc.html). If a LMS was located in the middle of a HUC, the HUC was split into two. Seventy-nine subwatersheds were located within the study area, with a mean drainage area of 63 km$^2$, minimum of 11 km$^2$, and maximum of 146 km$^2$.

## 2.4 Anthropogenic Factors

### 2.4.1 Land Uses

Land use variables were derived from the U.S. conterminous Wall-to-wall Anthropogenic Land use Trends (NWALT) dataset (Falcone 2015). We aggregated NWALT land use designations into three major categories: urban (including residential, transportation, industrial, and commercial development), agriculture (pasture and crop), and undeveloped (semi-developed, low use, and wetlands). Semi-developed land was included with undeveloped because it is mostly comprised of forested land in central NC (Miller et al., 2019). We further split urbanization constructed before and after a given date (e.g., 1980, 2000)

and between low and high density. In order to determine when urbanization occurred in the region, we interpolated available NWALT data (1974, 1982, 1992, 2002, and 2012) to obtain year-specific land use values for each subwatershed. Since our study extended beyond 2012, we also used linear extrapolation for years 2013-2017 based on 2002 and 2012 values. Land use trends throughout the study period were generally gradual, such that modest linear extrapolation was considered reasonable (Fig. 2).

### 2.4.2 Point Sources

Point sources included major (> 0.044 m$^3$/s) and minor wastewater treatment plants (WWTPs) (Fig. 1, Fig. S1, Table S1). Discharge data were obtained from NC DEQ and included monthly TN and flow values. However, many WWTPs had numerous missing months, so we determined annual loads as the product of yearly median concentrations and flows for each WWTP. LMSs with major WWTPs in their watersheds were only modeled starting in 1994 due to a lack of discharge data

before that year (i.e., HR1,3,5, NH1-3, and FL1,3,10), while LMSs without major WWTPs were modeled from 1982 depending on data availability. Only one LMS (HR4) with pre-1994 monitoring data included a minor WWTP in its watershed. Since the minor WWTP represented < 3% of the LMS mean load, we assumed the pre-1994 load was equal to the mean post-1994 load. TN trends of point source discharges aggregated by basin are shown in Fig. 2.

### 2.4.3 Livestock

The livestock in subwatersheds were estimated from county-level US Department of Agriculture (USDA) census and survey reports (https://www.nass.usda.gov/). Cow and hog data covered our entire study period (1982-2017) while chicken data were available every five years beginning in 1997. For missing years between census dates, chicken counts were interpolated,



whereas chicken counts before 1997 were assumed to be equal to 1997 values. Only two incremental watersheds (HR1, 3) had large chicken counts (> 1,000,000 and > 150,000, respectively) and these watersheds were not modeled before 1994 (as they

were also missing major WWTP discharge data).

To represent the locations of livestock throughout the region, county-level data were assigned to incremental watersheds based on an area ratio. Major urban areas were excluded when calculating these proportions, as livestock were assumed to be located outside cities. Livestock counts were then divided into the subwatersheds (using area ratios). However, chickens in Chatham County were accounted for differently because a majority of Chatham's chicken farms (>90%) are located outside of the JL

basin, and the county has a relatively high chicken count (>3,000,000; USDA). Chatham County records (*opendata-chathamncgis.opendata.arcgis.com*) were used to estimate that 8.2% of the county's chickens were within the JL basin. Basin level trends of livestock are shown in Fig. 2.

**2.5 Precipitation**

Monthly precipitation estimates for this study were obtained from the PRISM Climate Group

(http://www.prism.oregonstate.edu/). These data were processed using the R package "raster" (Hijmans et al., 2015; R Core Team, 2019) to determine mean annual precipitation for each subwatershed. There was substantial variation in precipitation among years (0.82 – 1.59 m/yr) and among different subwatersheds within the same year (Fig. S2).

**2.6 Nutrient Load Calculations**

Our model required yearly TN loadings at each LMS for Bayesian inference (i.e., calibration). Most riverine monitoring

programs measure streamflow daily, whereas nutrient concentrations are sampled less frequently (e.g., monthly). In this study, daily TN concentrations were estimated using the USGS Weighted Regressions on Time, Discharge, and Season (WRTDS; Hirsch et al., 2010). WRTDS develops a semi-parametric regression for each day in the estimation period where observations that are collected under similar conditions to the estimation date (in terms of time, discharge, and season) are more heavily weighted. For some LMS sites, there were rapid temporal changes in nutrient loading associated with WWTP plant upgrades.

In order to not bias WRTDS loading estimates near these events, the period of record was split into two (Table S2).

Some LMSs had incomplete monitoring data (daily flow or water quality samples) during our study period (Table 1). If a LMS was missing flow data for a given year, that year was omitted. However, nutrient loads could be estimated by WRTDS for years with missing water quality data. Gaps in water quality samples of up to one year were considered acceptable, as preliminary analysis (removing single years of observational data at random) showed that a one-year gap affected loading

estimates by less than 1%. In addition, at least 6 samples were required in both the beginning and ending year of each loading record.

Uncertainty in loading estimates were determined through subsampling of three NC stations that had nearly daily TN observations for at least 7 consecutive years (Strickling and Obenour, 2018). By comparing TN loads based on the full dataset





and different subsets, we estimated the coefficient of variation (and consequently the standard deviation) of WRTDS estimates
based on the number of water quality samples available for a given year (Fig. S3).

The response variable in our model was the change in nutrient load across an incremental watershed, defined as the difference between the load at an incremental watershed's downstream LMS and the cumulative load from any upstream LMSs. For sites with no upstream LMSs, the incremental load is equal to its total load. The uncertainties of incremental loads were calculated based on the relationship between correlated random variables (Eq. (1); Kottegoda & Rosso, 2008):

$$\tilde{\sigma}_{i,t}^2 = \sigma_{i,t}^2 - 2\sum_{k=1}^{n} \rho_{i,k}\sigma_{i,t}\sigma_{k,t} + \sum_{k=1}^{n}\sum_{l=1}^{n} \rho_{k,l}\sigma_{k,t}\sigma_{l,t} \tag{1}$$

Where $\tilde{\sigma}_{i,t}^2$ is the incremental load variance for a given incremental LMS watershed (i) in year (t), with $n$ upstream LMSs ($k$, $l$). Here, $\sigma_{i,t}^2$ is the error variance at the downstream LMS, $\sigma_{k,t}$ and $\sigma_{l,t}$ are the WRTDS standard deviations at upstream LMSs, and $\rho_{i,k}$ and $\rho_{k,l}$ are correlation coefficients between LMS loadings.

**2.7 Model Construction**

Our model is formulated similar to Strickling and Obenour (2018). Within a Bayesian framework, we relate deterministically predicted incremental loads ($\hat{y}_{i,t}$; Eq. (2)) to an inferred incremental load ($y_{i,t}$). The watershed-level random effect ($\alpha_i$; Gelman et al., 2014) accounts for spatial variability not explained by the deterministic prediction ($\hat{y}_{i,t}$), and the residual error (with standard deviation, $\sigma_\varepsilon$) primarily accounts for temporal variability unexplained by the deterministic prediction. The hyperdistribution of the normally distributed watershed-level random effect is centered on zero, with variance $\sigma_{LMS}^2$.

$$L(y_{i,t}) \sim N( L(\hat{y}_{i,t} + \alpha_i), \sigma_\varepsilon) \tag{2}$$
$$\alpha_i \sim N(0, \sigma_{LMS})$$

L(y) is the natural log transformation of $y + 10^5$ (kg/year of TN). This transformation reduces heteroscedasticity in residuals while accounting for any negative incremental loads that would produce non-real values when log transformed. Negative incremental loads are possible, especially for incremental watersheds with large impoundments that retain a substantial portion
of the load from upstream LMSs.

The inferred incremental load ($y_{i,t}$; Eq. (2)) is related to the WRTDS incremental estimates ($\tilde{y}_{i,t}$; Eq. (3)) by taking into account the uncertainty of those loading estimates ($\tilde{\sigma}_{i,t}$; Eq. (1)).

$$\tilde{y}_{i,t} \sim N(y_{i,t}, \tilde{\sigma}_{i,t}) \tag{3}$$

Within the model, the deterministic prediction ($\hat{y}_{i,t}$) is calculated by aggregating incremental watershed source contributions
and subtracting in-stream losses from upstream LMS loads (Eq. (4)).

$$\hat{y}_{i,t} = L_{i,t,ur1} + L_{i,t,ur2} + L_{i,t,ag} + L_{i,t,und} + L_{i,t,ps} + L_{i,t,ch} + L_{i,t,h} + L_{i,t,cw} - U_{i,t} * r_{i,z} + \varepsilon_{i,t} \tag{4}$$

Contributions are calculated for two urban ($L_{i,t,ur1}$; $L_{i,t,ur2}$), agricultural ($L_{i,t,ag}$), and undeveloped ($L_{i,t,und}$) lands, point sources ($L_{i,t,ps}$), chickens ($L_{i,t,ch}$), hogs ($L_{i,t,h}$), and cows ($L_{i,t,cw}$). Loads from upstream incremental watersheds ($U_{i,t}$) are reduced by their expected in-stream and reservoir losses ($r_{i,z}$; Eq. (6); see *Nitrogen Retention*). Each source-specific load is calculated as follows:





$L_{i,t,x} = \beta_x (\tilde{p}_{i,t}{}^{\gamma_x}) * \mathbf{a^T}_{i,t,x} * (1 - \mathbf{r}_{i,t})$             (5a) (land use)

$L_{i,t,x} = \beta_x (\tilde{p}_{i,t}{}^{\gamma_x}) * \mathbf{h^T}_{i,t,x} * (1 - \mathbf{r}_{i,t})$             (5b) (livestock)

$L_{i,t,x} = \beta_{ps} * \mathbf{w^T}_{i,t} * (1 - \mathbf{r}_{i,t})$             (5c) (point source)

where $L_{i,t,x}$ (kg/yr) represents the total contributed load for a given LMS (i), source (x), and year (t). Parameter $\beta_x$ represents a land or livestock export coefficient (kg/ha/yr or kg/an/yr) or the point source (i.e., WWTP) delivery coefficient (unitless, 0-1).

Parameter $\gamma_x$ is the precipitation impact coefficient (PIC, unitless) for a given nonpoint source, which is parameterized as a power relationship with the export coefficient (Eq. (5)). PICs differ by source, but are related to each other through a common hyperdistribution with mean $\mu_\gamma$ and standard deviation $\sigma_\gamma$ (Table S3). This formulation differs from the linear relationship between precipitation and loading used in Strickling and Obenour (2018) and avoids potentially negative loading values during extremely low flow years. Point sources do not have a PIC term (Eq.(5c)) as the WWTP data already account for yearly

variation. Scaled annual precipitation ($\tilde{p}_{i,t}$) for each incremental watershed is determined by dividing by the mean precipitation of the study area. Often, a given source type was distributed among multiple locations (e.g., subbasins) within an incremental watershed. To account for this, $\mathbf{a^T}_{i,t,x}$, $\mathbf{h^T}_{i,t,x}$, and $\mathbf{w^T}_{i,t,x}$ are transposed vectors of sources (i.e., ha of land use, counts of livestock, and load from WWTPs, respectively) across different locations that are multiplied by a vector ($\mathbf{r}_{i,t}$) of location-specific stream and reservoir retention losses.

## 200   2.8 Nutrient Retention

Nitrogen retention in streams is represented based on a first-order decay rate ($-\kappa$, d$^{-1}$) and mean stream residence time ($\tau_z$, d) for each path (z) from a given source (subwatershed or point source) to its downstream LMS. Estimated mean stream velocities are from NHD+ (Moore and Dewald, 2016). Travel distance was estimated as half the distance of the longest flow path within the source subwatershed plus the distance from the subwatershed to the downstream LMS. Nitrogen retention in reservoirs is

modeled as a function of hydraulic loading rates (ratio of flow to surface area, $q_w$, m/yr) for each waterbody (w) and a mass transfer coefficient, ($\omega$; m/yr.; Kelly, 1987). An overall retention rate ($r_{t,z}$), combining streams and reservoirs, for each path (z) and year is determined as:

$$r_{t,z} = 1 - \exp\left(-\kappa * \tau'_{t,z}\right) * \prod_w \exp\left(\frac{-\omega}{q'_{t,w}}\right) \qquad (6)$$

allowing for multiple waterbodies along each flow path (i.e., the product function). While Strickling and Obenour (2018) used

constant retention rates, here we allow rates to vary interannually by relating travel time and hydraulic loading to annual precipitation. Specifically, annual stream travel times ($\tau'_{t,z}$) and reservoir hydraulic loading rates ($q'_{t,w}$) are determined based on a retention PIC ($\gamma_{ret}$) and normalized yearly precipitation ($p_t$; yearly precipitation minus mean precipitation divided by standard deviation), specific to each incremental watershed and year (Eq. (7a,b)):

$$\tau'_{t,z} = \frac{\tau_z}{(1 + \gamma_{,ret} * p_t)} \qquad (7a)$$

$$q'_{t,w} = q_w * (1 + \gamma_{ret} * p_t) \qquad (7b)$$



## 2.9 Bayesian Inference

All model parameters were assigned a prior probability distribution (Table S3). Informative priors were used when previous studies reporting similar parameters were available. Prior distributions for land export rates were taken from Dodd (1992), while stream retention rates were adapted from previous SPARROW studies (Hoos and McHahon, 2009; Garcia et al., 2011).

Prior distributions for chicken and hog TN export coefficients were adapted from Strickling and Obenour (2018) to represent kilograms of TN per animal per year. Essentially uninformative priors (i.e., wide uniform priors) were used for the remaining parameters.

For comparison, models were calibrated with urban lands split in four different ways: 1) pre and post-1980 urban lands, 2) pre and post-2000 urban lands, 3) low versus high density urban (high-density residential only), and 4) low vs. high density urban

(high-density residential, industrial, and commercial). In order to evaluate the best representation, we compared model fit and the degree of overlap between the marginal posterior distributions of the two different urban export coefficients within each model.

Models were parameterized within the Bayesian framework using RStan software in R (R Core Team, 2019; Stan Development Team, 2020). RStan uses Hamiltonian Monte Carlo sampling of the posterior distribution and often converges faster than other

samplers (Gelman et al., 2015). Three parallel chains of 20,000 iterations with a burn-in period of 5,000 iterations (that were discarded), creating 9,000 posterior samples after thinning (accepting every fifth iteration). Parameters were considered to have converged when their scale reduction coefficient ($\hat{R}$) was below 1.1 (Gelman and Rubin, 1992).

## 2.10 Model Assessment and Validation

Predictive performance was assessed using the coefficient of determination (i.e., variance explained, $R^2$, Faraway, 2016) for

incremental nutrient loads. Predicted incremental loading estimates ($\hat{y}_{i,t}$) were derived using the Bayesian mean posterior values and compared to WRTDS loading estimates ($\tilde{y}_{i,t}$). Model performance was assessed for LMSs in the HR, NH, and FL watersheds with and without their watershed-level random effects. To test the ability of the model to make out-of-sample predictions, we performed a 3-fold cross-validation (Elsner and Schmertmann, 1994). The data were split into three groups by major watershed (HR, NH, and FL), and the model was trained on 2 of the 3 watersheds, in turn. Predictions were then made

on the excluded major watershed (in turn).

## 3 Results

### 3.1 In-stream Nutrient Loading

WRTDS-derived annual loading estimates are quite noisy (Fig. 3; Fig. S4) due largely to hydrologic variability, while flow-normalized estimates help illustrate long-term trends (Hirsch et al., 2010). TN loads in all basins decreased substantially from

1980 to the late 1990s. However, post-2000 loading patterns are inconsistent both within and across basins. In the HR


watershed, TN loading has steadily increased since 2000. In the NH watershed, substantial increases in loads were seen after 1995, though subsequent WWTP improvements reduced loading in some tributaries. In FL, large post-2000 TN reductions occurred in FL1 and FL10, both of which have major WWTP discharges, while loadings in other FL watersheds have remained constant or trended upwards.

### 3.2 Comparing Different Urban Land Classifications

The hybrid watershed model explains the spatial and temporal variability of in-stream (WRTDS-derived) TN loads based on precipitation and nutrient source distributions. To explore urban TN sources in more detail, we compare model variations with different classifications of urban land use considering the age, density, and type of urban development (Table 2). We find that a classification based on a pre/post 1980 split results in significantly different export rates, with the pre-1980 urban lands exporting more than twice the amount of post-1980 urban lands. In addition, the pre/post 1980 division leads to the highest $R^2$ values for individual LMSs. None of the other urban splits result in significantly different parameter estimates at a 95% credible level. However, export rates from high-density and older urbanization are consistently higher than the less dense and newer urban lands. Among the two splits based on density, combining high-density residential, industrial, and commercial lands has higher predictive power than just separating high-density residential from other urbanization ($R^2$= 0.47 vs. 0.44; Table 2).

### 3.3 Model Posterior Parameter Estimates

The posterior ECs ($\beta_x$) of the preferred model (with the pre/post 1980 urban split) show that urban and agricultural lands both contribute substantial TN per unit area (Table 3; Fig. 4). In particular, pre-1980 urban development exports 9.4 kilograms per hectare per year (kg/ha/yr) of TN, while post-1980 development exports 3.9 kg/ha/yr, though with a relatively wide credible interval. Agriculture also contributes a substantial 4.0 kg/ha/yr, while undeveloped lands export a relatively low 0.7 kg/ha/yr. Model posterior distributions indicate that parameter uncertainties are reduced substantially relative to the prior distributions (Fig. 4). In addition to pre-1980 urban export being significantly greater than other forms of development (at 95% credible level), undeveloped land has significantly less export than all developed lands (Table 3).

Land use ECs represent expected nutrient export for a year with mean annual precipitation (i.e., $\tilde{p}_{i,t} = 1$; Eq. (5a)). Because the relationship between export and precipitation is nonlinear, the export coefficients represent median (but not mean) loading rates. The precipitation impact coefficients (PICs) can be used to calculate TN export during low and high flow years. Agriculture has the largest PIC (4.0; Table 3) implying that export from agricultural lands (crop and pasture) vary the most due to rainfall. During a high flow year (90$^{th}$ percentile $\tilde{p}_{i,t}$=1.18), nutrient export for agriculture would almost double from 4.0 to 7.7 kg/ha/yr. For a low flow year (10$^{th}$ percentile $\tilde{p}_{i,t}$=0.81), nutrient export for agriculture (1.7 kg/ha/yr) is less than half the median export. Pre-1980 urban lands showed the lowest variation due to precipitation ranging from 81% of normal export in low flow years up to 122% in high flow years.




## 3.4 Spatial Variation in Nutrient Export and Retention

The TN export from nonpoint sources (i.e., land use and livestock) was calculated for each subwatershed (Fig. 5a) using mean precipitation, and mean posterior land and livestock ECs ($\beta_{ec}$; Table 3). Since the most intensive nutrient export comes from pre-1980 urban lands, subwatersheds intersecting the urban cores of major cities (Fig. 5a) have the largest expected export. Predominantly rural watersheds export between 1-3 kg/ha/yr of TN, while urban cores export over 6 kg/ha/yr.

On average, 13% of TN is retained within the JL and FL stream and waterbody networks (Fig. 5b). Little TN is retained in subwatersheds close to JL and FL and along higher order streams, while large TN removal rates (> 70%) occur for subwatersheds located upstream of reservoirs in the upper northwest portions of the JL basin. Overall, more TN retention occurs in reservoirs than in streams. Residence times and hydraulic loading rates are also affected by precipitation as modulated by the PIC for stream retention ($\gamma_{ret}$; 0.07; Table 3). For one standard deviation increase in yearly precipitation (17.1 cm), expected stream residence times and hydraulic loading rates decrease roughly 7% (Table 3; Eq. (7a,b)). During low precipitation years (lower 33%), 15% of TN is retained in the JL and FL networks, while 12% is retained during normal and high precipitation years (upper 67%).

Watershed-level random effects account for unexplained spatial variations in nutrient loading. For example, the negative random effects for small watersheds comprised of mostly pre-1980 urban development (NH7, NH8, FL2) imply these watersheds export less TN (-2.5, -1.8, and -1.2 kg/ha/yr, respectively; Fig. S5) than typical pre-1980 urban lands (9.4 kg/ha/yr; Table 3). Similarly, two LMSs (FL6, FL9) located directly downstream of large impoundments had negative random effects implying these impoundments may be particularly efficient at trapping nutrients. On the other hand, three watersheds (NH1, HR5, and FL1) located just downstream of major WWTP discharges have elevated TN watershed-level effects, suggesting loads may be underestimated by the available point source data (Fig. S5).

Watershed random effects were also compared to regional soil distributions (Fig. S6) since soil and geologic properties have been linked to both nutrient loading and transport (Preston et al., 2011). Triassic soils, in particular, have a distinct geologic history (NC Geol. Surv, 2019) with lower infiltration rates that result in higher erosion potential and lower baseflows in streams (Tetra Tech 2014). Six LMS watersheds had predominantly Triassic soils but random effects associated with these watersheds are inconsistent in sign and magnitude (Fig. S5). Thus, soil conditions do not appear to play a major role in determining nitrogen export in our study area.

## 3.5 Nutrient Source Allocations Over Time

Yearly TN loadings from 1994-2017 were calculated based on mean model parameters, land use, livestock counts, and precipitation. Only the NH watershed shows a clear downward trend in TN loading, which appears to be largely driven by WWTP discharge reductions (Fig. 6). In both the HR and FL watersheds, annual loadings nearly tripled from the lowest to highest precipitation years due to high levels of agricultural lands which substantially increase export during wet years (Fig.





6). This high-level of interannual variation, makes it difficult to distinguish any positive or negative trends in these basins over the study period. Additionally, model residuals were plotted against time with no noteworthy trends (Fig. S7).

### 3.6 Model Skill Assessment

The full hybrid model, including random effects, explains 95% of the variation in the WRTDS loading estimates at LMSs (Fig. S8). Discounting the random effects, the model still explains 93% of TN loading variability. The model (with watershed-level random effects) explains 96% of the variation in the HR, 92% in NH, and 83% in FL (Fig. S8), suggesting some spatial variability in model performance. In cross validation, the predictive ability of the model remained high, with the $R^2$ of the full TN model (without watershed-level random effects) lowered slightly from 93% to 90%. $R^2$ was also tabulated for individual

LMSs to assess the model's ability to predict the temporal variability of loadings for a specific LMS. $R^2$ ranged greatly from below 0 (FL2, JL2, JL7) to above 0.80 (JL4, FL7, FL10) with a mean $R^2$ being 0.48 (as in Table 2).

## 4 Discussion

### 4.1 Nutrient Export Rates and Discharge Coefficients

In this study, we aim to enhance our understanding of nitrogen export, especially as it relates to land use and different types

of urbanization. Variation in urban TN export has often been associated directly with population density (Bales et al., 1999; Burns et al., 2005; Line, 2013; Tetra Tech, 2014) or with proxies for density like net food imports (Hong et al., 2011; Sinha and Michalak, 2016). In this study, we compare variations in urban export due to the age of the urbanization versus different urban land covers (e.g., high and low-density residential, commercial). Pre-1980 urban lands and high-density residential are moderately correlated in our study area ($r^2$=0.64), yet Bayesian posterior parameter estimates show that export from low and

high density urban areas were not statistically different from each other. However, TN export from pre and post-1980 urban areas are statistically significantly different (Table 3). This suggests that urban infrastructure age and historical development practices are more important than population density.

Various mechanisms, beyond density, could explain why older urban areas export more TN than recently constructed urban areas. Increased impervious connectivity in pre-1980 urban areas may lead to elevated runoff and nutrient washoff (Wolheim

et al., 2005). In addition, pre-1980 urban development generally lacked stormwater management and erosion control measures (Howells, 1990; NC DEMLR, 2019). Therefore, streams in these areas often exhibit urban stream syndrome with elevated banks, eroded channels, and low biological health (Paul and Meyer, 2001; Bernhardt and Palmer, 2007; Miller et al., 2019). Older neighborhoods are also likely to have larger trees and thus more leaf litter over impervious surfaces, which can also increase nutrient export (Janke et al. 2017). In addition, leaky sewer infrastructure in pre-1980 urban areas might be a

substantial source of nutrients (Kaushal et al., 2011; Pennino et al., 2016) as compared to newer and more reliable infrastructure in post-1980 urban areas.





Model results indicate that post-1980 urban and agricultural lands exported similar amounts of nitrogen (3.9 and 4.0 kg/ha/yr, respectively; Table 3). Our estimated agricultural export rate is lower than previous studies (Dodd et al. 1992; Strickling and Obenour 2018), which may be related to the fact that over 90% of agricultural lands in our study area are pasturelands, rather

than croplands (Falcone 2015). Post-1980 urban export shows the most uncertainty in model posteriors (Table 3; Fig. 4). This might be due to the inconsistency of regulations being applied both temporally and spatially from 1980 to the present, and it might also indicate variation in the best management practices (BMPs) used in the region. Finally, undeveloped lands have very low export (0.7 kg/ha/yr; Table 3) with moderate uncertainty (0.1-1.5 95% interval; Table 3). This mean value is roughly three times lower than previous studies in the region (Tetra Tech 2014; Strickling and Obenour 2018).

Livestock export coefficients for chickens, hogs, and cows (0.01, 0.04, and 0.50 kg/yr, respectively) imply that less than 1% of the TN produced by these animals (i.e., 0.6, 9.9, and 54.8 kg/an/yr, respectively; Ruddy et al., 2006) results in excess TN pollution. Thus, livestock-related nutrient export appears to account for <2% of nutrient loading in our study area (Fig. 6; Table S4). At the same time, it is important to note that livestock waste that is used to replace other (e.g., synthetic) fertilizers is generally represented by the agricultural land export term.

Our point source coefficient discounts WWTP loads by nearly 20% ($\beta_{ps}$=0.83; Table 3). One potential explanation for this result is that TN from WWTP (primarily nitrate) is processed and retained in stream networks more efficiently than TN from nonpoint sources. For example, increased denitrification rates have been observed downstream of WWTPs due to altered biochemical conditions (Wakelin et al. 2008; Rahm et al. 2016).

### 4.2 Inter-Annual Variability

Our analysis of interannual variability was facilitated through three enhancements to the hybrid watershed modeling approach of Strickling and Obenour (2018). First, we allow for retention to vary across years due to changes in hydrology, represented parsimoniously by a PIC for stream residence times and reservoir hydraulic loading rates ($\gamma_{,ret}$; Eq. (7a,b)). This modification allows in-stream nitrogen retention to vary interannually (~ 8% variation for 1 SD in annual precipitation). Consequently, stream and reservoir retention rates ($\kappa$, $\omega$; Table 3) represent retention during mean precipitation years. In this region, the

majority of retention in the stream network occurs in reservoirs, as opposed to in streams. Our mean stream retention rates are comparable in magnitude (0.04 d$^{-1}$; Table 3) to regional hybrid models (Strickling and Obenour, 2018; Gurley et al., 2019), but lower than previous watershed process models (Tetra Tech, 2014). Accurately quantifying stream retention rates is important to operators of WWTPs and regulators in order to accurately determine nutrient offsets for projects, which are often valued in millions of dollars (personal communication Jim Hawhee, NC DEQ, 2020).

Second, we modeled the effects of precipitation variability on land use export using power functions ($\gamma_{,x}$; Eq. (5)) instead of linear relationships (Strickling and Obenour, 2018). The new formulation recognizes that as precipitation increases, its marginal effect on nutrient loading may intensify, as infiltration and evapo-transpiration rates are exceeded (Chin, 2013). Consistent with this explanation, agriculture has the highest PIC (4.0; Table 3), indicating that when annual precipitation is 20% higher than the mean, TN export from agricultural lands will approximately double. On the other hand, pre-1980 urban





lands, which are substantially impervious, have the lowest PIC values and only export 24% more TN for a 20% increase in annual precipitation. Third, we used a higher spatial resolution monitoring network than Strickling & Obenour (mean incremental watershed of 321 km$^2$ vs. 1535 km$^2$; 2018) which allowed us to account for export in both small and large watersheds.

Finally, we compiled all nitrogen loadings for 20+ years throughout the JL and FL basins (1994- 2017; Fig. 6) to analyze 375 trends. Inter-annual variation was large (~ 3x from lowest to highest year) in both HR and FL, yet NH loadings showed relatively little variation. This was due to high levels of agricultural lands in HR and FL, which have the highest PIC values. NH loadings, in contrast, were dominated by pre-1980 urbanization (lowest PIC) and point sources (no PIC). By separating TN loadings into their sources, we were able to identify trends in specific nitrogen sources that could help inform management of individual watersheds. For example, though loadings in the NH watershed declined over the study period from 6 x 10$^5$ kg/yr 380 to 4 x 10$^5$ kg/yr, this overall trend was driven by a nearly 50% reduction of point source discharges (4 x 10$^5$ down to 2.1 x 10$^5$ kg/yr). At the same time, load attributable to post-1980 urbanization increased almost two-fold, from 0.2 x 10$^5$ up to 0.4 x 10$^5$ kg/yr (Fig. 6).

## 4.3 Potential Nutrient Reductions

Model results strongly indicate that the majority of nutrient inputs to JL and FL are from anthropogenic sources. Based on 385 identified sources of TN in the watershed, four management strategies would potentially lead to large nutrient load reductions: 1) reduction of point source loadings (i.e., WWTPs), which remain the largest individual source of TN, 2) retrofitting or replacing infrastructure in older urban environments (i.e. pre-1980 urban), which are the largest nonpoint source of TN per unit area, 3) mitigating TN loading from agricultural lands, especially during wet conditions, and 4) limiting, reducing, or offsetting the removal of undeveloped land, which has the lowest export rate.

The effectiveness of these nutrient control strategies will vary across different hydrologic conditions. For example, point sources are responsible for 44% of TN loadings to JL, and 14% to FL from 1994 to 2017. However, these percentages rise to 55% and 24%, respectively, during dry years (Table S4). On the other hand, agricultural TN export accounts for 18% of JL loadings and 30% of FL loadings during normal flow years, but that number increases to 27% and 37% during high flow years (Table S4). Therefore, strategies aimed at mitigating point source loadings will affect lake loadings more in low flow years, 395 while agricultural strategies will have a larger effect in high flow years.

Undeveloped lands have the lowest nutrient export of all land-use sources (Table 3), which is consistent with findings from targeted water quality monitoring recently done in JL (NC Policy Collaboratory 2019). However, undeveloped areas are decreasing in this region (1.5 km$^2$/yr, 3 km$^2$/yr, and 2 km$^2$/yr in HR, NH, and FL basins, respectively, from 1995 to 2015). Even though more recent development (post-1980) has significantly lower TN export when compared to pre-1980 400 development, it still exports approximately six times more TN than undeveloped lands (3.9 vs. 0.7 kg/ha/yr; Table 3). At the same time, post-1980 urban export rates are similar to agricultural rates, implying that new urban construction in agricultural areas may have limited impact on total nutrient loading.





### 4.4 Summary and Future Directions

To efficiently manage watershed nutrient loading, it is critical to identify the major sources and locations of nutrient loading
under differing hydroclimatological conditions. We enhanced and applied a "hybrid" watershed modeling approach within a
Bayesian framework to characterize TN loading rates from point and nonpoint sources and tested different classifications of
urban land. By modeling interannual variability, the model provides an assessment of how land use change and
hydroclimatological variations have affected nutrient loading over time. Process-based and hybrid modeling approaches
(e.g., SPARROW; Gurley et al. 2019) have been widely used to determine nitrogen loading rates, but these applications are
often limited by an inability to capture interannual variations in loading and/or provide holistic parameter calibration with
uncertainty quantification. Compared to previous Bayesian hybrid watershed modeling studies (Qian et al., 2005; Wellen et
al., 2012; Strickling & Obenour, 2018), this study advances our ability to account for interannual variability in export and
retention. The study also discriminates how export rates vary across a relatively large number of source types (4 land uses, 3
livestock types, and point sources). Our ability to resolve 18 process-based parameters within the Bayesian framework is
facilitated, in part, by a relatively dense stream monitoring network and modern tools for Bayesian inference (Monnahan et
al., 2017).

In this study, we identify areas of elevated TN export. In particular, we find that pre-1980 urban areas are hot spots for nonpoint
TN loading. In addition, watershed-level random effects help identify outlier watersheds that export significantly more or less
TN than the source distribution data would otherwise imply. Great costs have been incurred to protect waterways in the last
30-40 years without a clear understanding of how effective current policies have been in reducing nutrient loading (Parr et al.,
2016; Utz et al., 2016). Our results suggest that post-1980 construction and land development BMPs have helped to reduce
TN loadings from the built environment. We hope these findings will stimulate further research into the specific mechanisms
that result in lower TN export from newer development. Enhancing the hybrid model with BMP and wastewater infrastructure
data, in addition to more detailed land use and hydrography data, could be one approach for refining our understanding of how
specific development practices influence watershed-scale nutrient loading. Given that even newer (post-1980) development is
found to increase TN export by a factor of five or more, relative to undeveloped lands, further efforts are required to understand
and mitigate the adverse impacts of urban development on nutrient loading.



**Code/Data Availability**

Hydrometeorological and water quality data can be obtained from the public sources described in the methods (e.g., USGS,
Water Quality Portal). A complete set of compiled input data can be provided upon request to the authors
(jwmille7@ncsu.edu).

**Author Contribution**

Jonathan Miller and Daniel R. Obenour designed the research. Jonathan Miller and Kimia Karimi developed the codes and
created tables and figures. All authors analyzed the research. Jonathan Miller and Daniel R. Obenour prepared the manuscript
with contributions from all co-authors.

**Competing interest**

The authors declare that they have no conflict of interest.

**Acknowledgements**

We would like to thank Hayden Strickling for assistance with preliminary modelling files. We would also like to thank Bridget
Munger (NC DEQ) for data assistance with WWTP data.

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









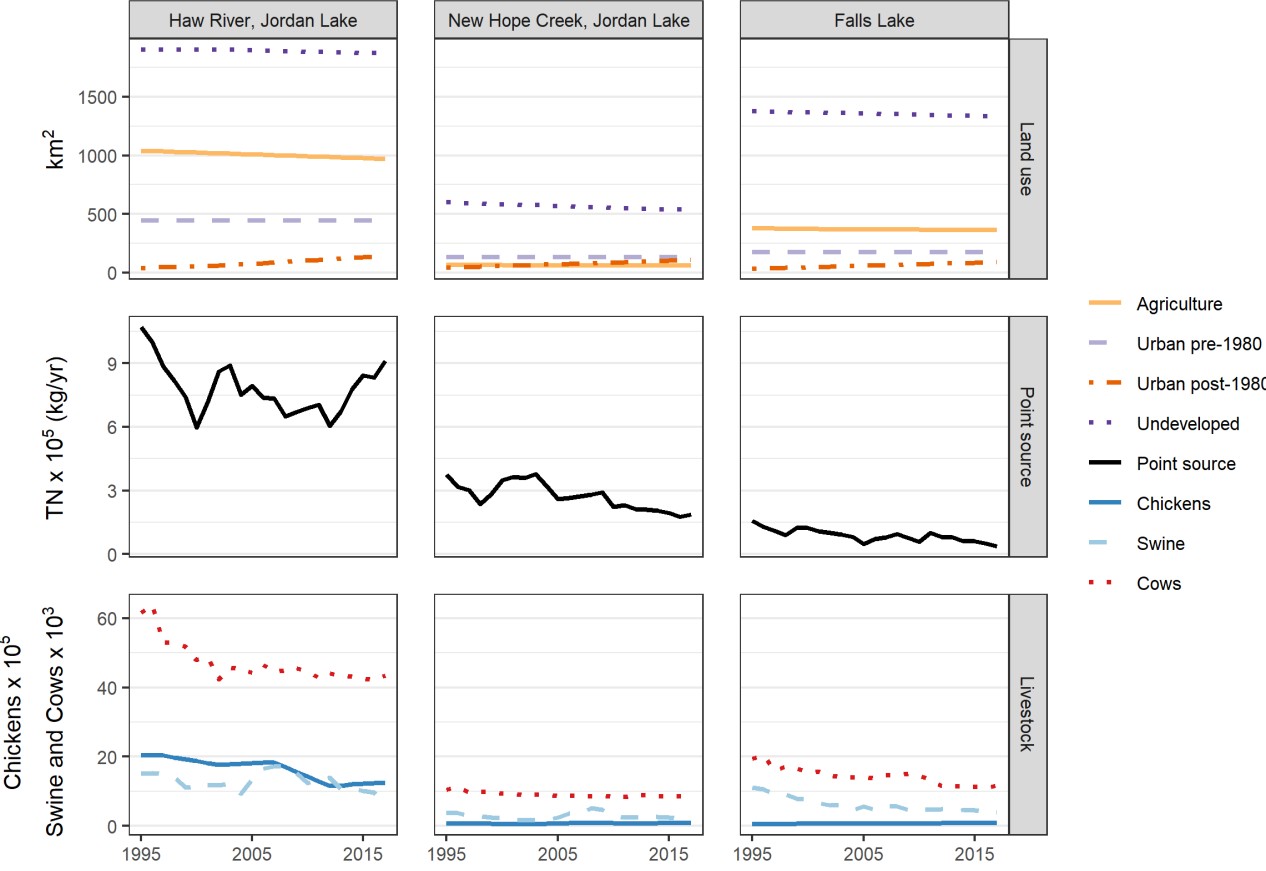

**Figure 2: Land use, point sources, and livestock trends from 1994-2017 in the Haw River, New Hope, and Falls Lake basins.**






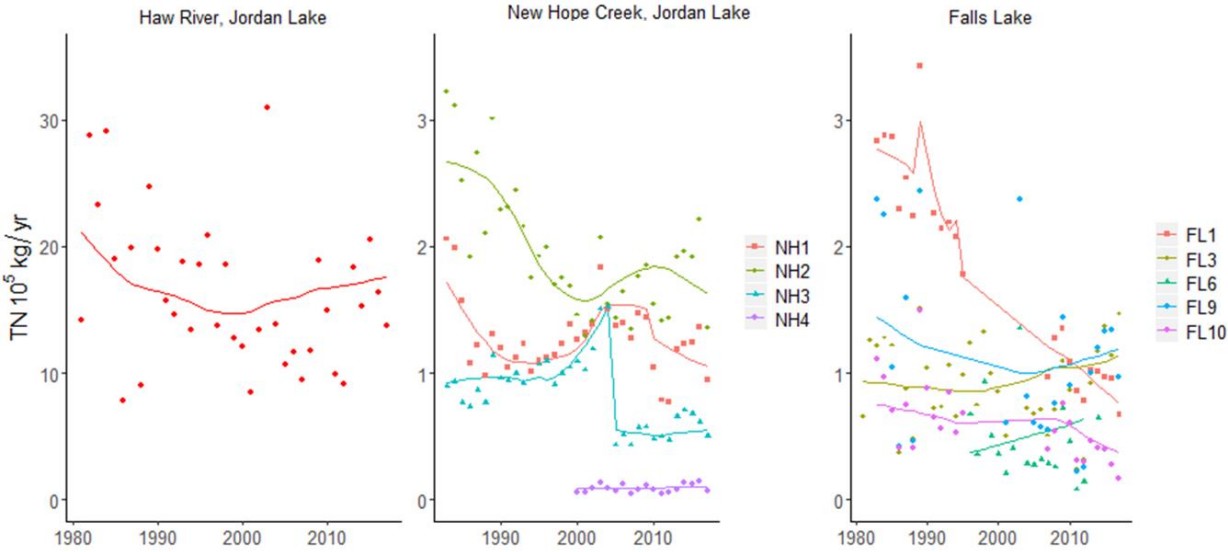

**Figure 3: Weighted-regression on Time, Discharge, and Season (WRTDS) annual nutrient loading estimates (points) and flow-normalized estimates (lines) for TN. For clarity, results are only shown for the most downstream load monitoring site (LMS) of each tributary to Jordan and Falls Lake. The LMS for Haw River is HR1. WRTDS loading estimates for other LMSs are provided in supporting information (Fig. S4).**






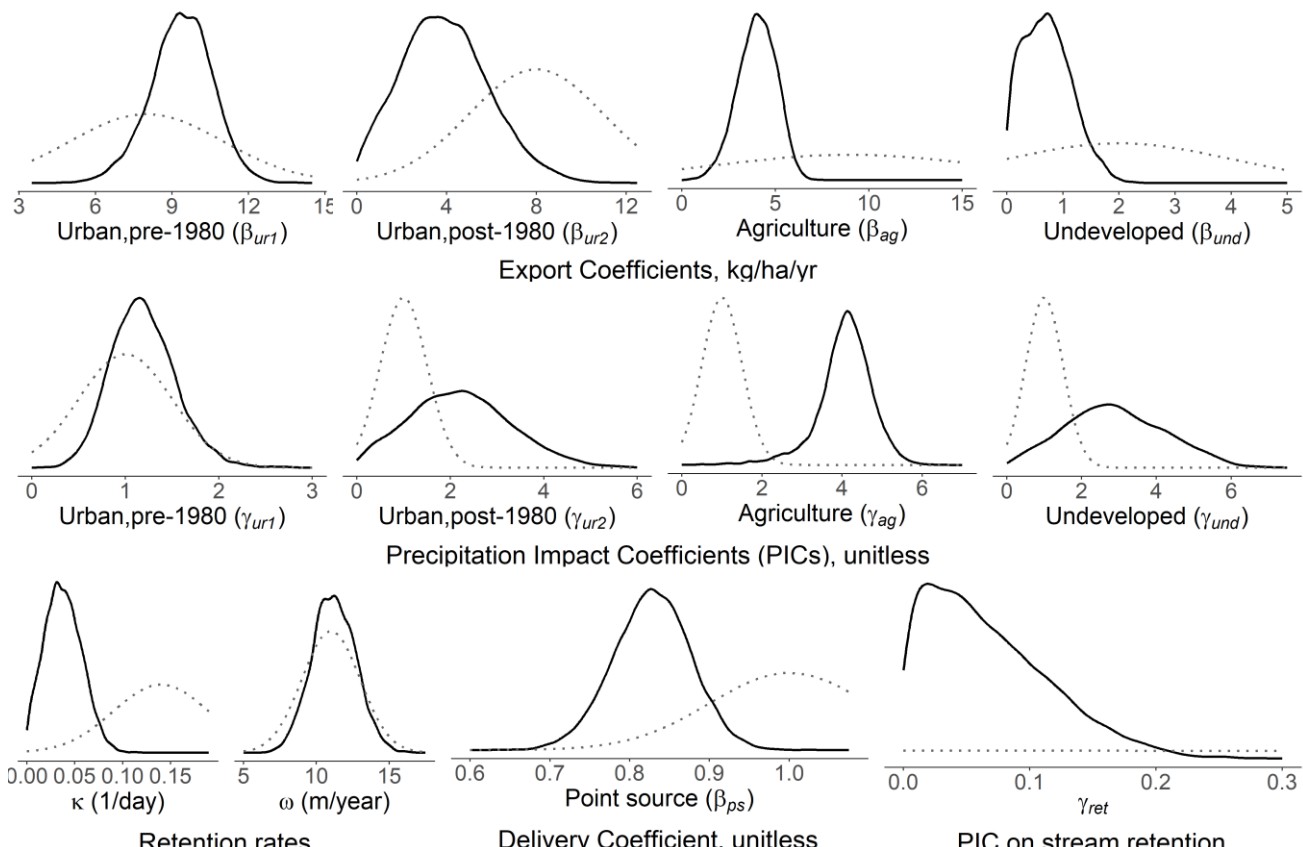

**600**    **Figure 4: Prior (dotted lines) and posterior (solid lines) distributions for selected model parameters. Note that priors and posteriors are provided for all parameters in Tables 4 and S3.**





**Figure 5: TN export (A) from land use and livestock by subwatershed; fraction of TN export from each subwatershed that is retained (B) in streams and reservoirs prior to reaching Jordan and Falls Lakes. Point source loads are shown separately as dots.**





**Figure 6: Total nitrogen export by year and major watershed separated by source. The star (\*) represents the total TN load that reached Jordan or Falls Lake.**




**Table 1: Load monitoring stations (LMSs) located in the Jordan (JL) and Falls Lake (FL) basins along with their drainage areas. LMSs belong to either New Hope Creek (NH) or Haw River basins of JL or FL. Years of record corresponds to time that loadings could be estimated (i.e., when daily flow and monthly water quality sampling was performed). The number of total nitrogen (TN) samples available is also shown.**

| LMS | Name | Res | Drainage area ($km^2$) | Years of record | # TN samples |
|------|------|-----|-------|-------|-------|
| NH1 | Morgan Creek, Jordan Lake | JL | 121.4 | 1994-2017 | 578 |
| NH2 | New Hope Creek | JL | 203.9 | 1994-2017 | 575 |
| NH3 | Northeast Creek | JL | 53.6 | 1996-2017 | 430 |
| NH4 | White Oak Creek | JL | 31.1 | 2000-2017 | 106 |
| NH5 | Morgan Creek, White Cross | JL | 21.4 | 2000-2017 | 116 |
| NH6 | Morgan Creek, Chapel Hill | JL | 103.2 | 2001-2013 | 141 |
| NH7 | Sandy Creek, Cornwallis | JL | 12.1 | 2009-2017 | 133 |
| NH8 | Third Fork Creek | JL | 41.2 | 2009-2017 | 107 |
| HR1 | Haw River, Bynum | JL | 3296.4 | 1994-2017 | 590 |
| HR2 | Cane Creek | JL | 19.6 | 1989-2017 | 227 |
| HR3 | Haw River, Burlington | JL | 1562.1 | 1994-2017 | 268 |
| HR4 | Reedy Fork , Gibsonville | JL | 316.6 | 1981-1986 2001-2017 | 341 |
| HR5 | N. Buffalo Creek | JL | 96.2 | 1999-2017 | 394 |
| HR6 | S. Buffalo Creek | JL | 88.6 | 2000-2017 | 343 |
| HR7 | Reedy Fork, Oak Ridge | JL | 53.4 | 2001-2017 | 255 |
| FL1 | Ellerbe Creek, Gorman | FL | 54.8 | 2006-2017 | 280 |
| FL2 | Ellerbe Creek, Murray | FL | 11.2 | 2009-2013 | 100 |
| FL3 | Eno River, Durham | FL | 367.2 | 1994-2000 2004-2017 | 375 |
| FL4 | Eno River, Hillsborough | FL | 171.0 | 1990-2017 | 223 |
| FL5 | Little River, Orange Factory | FL | 202.7 | 1988-2000 2005-2017 | 381 |
| FL6 | Little River, Fairntosh | FL | 246.4 | 1996-2011 | 196 |
| FL7 | Mountain Creek | FL | 20.8 | 1995-2011 | 156 |
| FL8 | Flat River, Bahama | FL | 385.9 | 1981-2011 | 472 |
| FL9 | Flat River, Dam | FL | 434.4 | 1983-1990 2003-2017 | 225 |
| FL10 | Knap of Reeds Creek | FL | 111.4 | 2006-2017 | 142 |






**Table 2: Posterior distributions of urban export coefficients (EC) split by age and development density. low-density (LD) vs. high-density (HD) urbanization. The probability (P) that older urban lands (or HD urbanization) exports more nitrogen than other urban lands was calculated by comparing Bayesian posterior draws. R2 represents the ability of the model to predict temporal variability of loading at each LMS. Mean R2 was determined by averaging the R2 of all 25 LMSs. HD vs. LD (1) compares high-density residential vs. other urban lands while HD vs. LD (2) defines HD as high-density residential, industrial, and commercial lands.**


| Case | Export Coefficient (EC) | Mean (TN/ha/yr) | 95% CI | P(EC1 > EC2) | Mean $R^2$ |
|---|---|---|---|---|---|
| A | 1. Pre 1980 urban | 9.4 | 7.3-11.3 | 98% | 0.476 |
|   | 2. Post 1980 urban | 3.9 | 0.9-7.3 |  |  |
| B | 1. Pre 2000 urban | 8.1 | 6.0-9.9 | 71% | 0.471 |
|   | 2. Post 2000 urban | 6.5 | 2.3-10.8 |  |  |
| C | 1. HD residential | 8.2 | 4.5-12.0 | 57% | 0.439 |
|   | 2. Other urban | 7.6 | 4.9-10.1 |  |  |
| D | 1. All HD urban | 7.9 | 5.6-10.0 | 72% | 0.468 |
|   | 2. Other urban | 6.5 | 3.0-10.0 |  |  |





**Table 3: Mean posterior parameter estimates for export and delivery coefficients (β; EC, DC), retention rates (κ, ω), and precipitation impact coefficients (γ; PIC) along with 95% credible intervals (CI).**

| EC, DC, and retention | | | PIC | | |
|---|---|---|---|---|---|
| Parameter | Mean | 95% CI | Parameter | Mean | 95% CI |
| $\beta_{ag}$ | 4.0 | 2.3-5.6 | $\gamma_a$ | 4.0 | 2.8-5.1 |
| $\beta_{ur1}$ | 9.4 | 7.3-11.3 | $\gamma_{ur1}$ | 1.2 | 0.7-1.8 |
| $\beta_{ur2}$ | 3.9 | 0.9-7.3 | $\gamma_{ur2}$ | 2.2 | 0.5-4.1 |
| $\beta_{und}$ | 0.7 | 0.1-1.5 | $\gamma_{und}$ | 2.9 | 0.8-5.2 |
| $\beta_{ch}$ | 0.01 | 0-0.02 | $\gamma_{ch}$ | 2.0 | 0.4-3.9 |
| $\beta_h$ | 0.04 | 0.01-0.07 | $\gamma_h$ | 2.0 | 0.3-3.8 |
| $\beta_{cw}$ | 0.52 | 0.06-0.95 | $\gamma_{cw}$ | 1.9 | 0.3-3.8 |
| $\beta_{ps}$ | 0.83 | 0.73-0.92 | $\gamma_{ret}$ | 0.07 | 0-0.16 |
| $\kappa$ | 0.04 | 0.01-0.07 | $\mu_\gamma$ | 1.8 | 1.1-2.4 |
| $\omega$ | 11.2 | 8.7-13.7 | $\sigma_\gamma$ | 1.1 | 0.7-1.6 |
| $\sigma_\varepsilon$ | 0.07 | 0.07-0.08 | | | |
| $\sigma_{LMS}$ | 1.34 | 0.90-1.91 | | | |