# Peer review of "Assessing interannual variability in nitrogen sourcing and retention through hybrid Bayesian watershed modeling"

_Hydrology and Earth System Sciences, 2021_

## Author Comment (AC1)

**Article:** Assessing inter-annual variability in nitrogen sourcing and retention through hybrid Bayesian watershed modeling

Response to Anonymous Referee #1: Responses in red

**General comments:**

This is a well-written paper, presenting an interesting model of nitrogen loading across river basins, accounting for temporal variability. In general, the methods appear to be appropriate, with assumptions and potential biases considered and appropriately accounted for, while the results are well interpreted and implications for policy are discussed.

I suggest below some specific comments, most of which are very minor in nature.

Thank you for the feedback on our manuscript; we address your specific comments below.

**Specific comments:**

Equation 1 (line 160). I am not sure I completely understand this formulation. It seems like there are only 2 upstream LMSs considered (k and l), while line 161 mentions "n". I do not have access to the original reference, but I wonder if some minor clarification would be helpful here.

"n" is a count variable that ranges from 1 to n. In our study, the largest value for n was 3 for site HR3. The "k and l" in the parentheses in this sentence confused this. We will clarify this in our revisions.

Section 2.9 (lines 216 to 232). Has a sensitivity analysis been carried out to investigate the effects of changing the informative priors? If not, I think this would be useful in understanding the robustness of the model. In any case, I think that some discussion of this is required.

In Strickling & Obenour (2018), such a sensitivity analysis was carried out on the data by running the hybrid watershed model with uninformative priors. This produced only small changes in the parameter estimates. In this study we are working with a larger observational dataset (25 load monitoring sites, compared to 21 sites in Strickling & Obenour), such that the influence of the priors is likely smaller. As such, we don't think this exercise would add new insights. We will note this briefly in our revisions to the Methods section.

Section 3.3 (lines 260 to 275). Can the CI endpoints be reproduced here? Currently, I feel that the point estimates without this context suggest greater certainty in these values than is the reality.

We agree that the point estimates might imply higher certainty than is warranted, but we do not want to duplicate all of the 95% CIs from Table 3, which might make the text cumbersome. We will add coefficients of variation (CV) in this section to convey the uncertainty in the point estimates.

**Technical corrections:**

Line 46. A comma between "reservoirs" and "using" might be useful.

We agree and will edit the text at line 46.

Line 144. I think "plant" is unnecessary, being effectively a repetition here.

We agree and will edit the text at line 144.

Line 172. Is $10^5$ definitely correct here? (It seems very large for an offset for a log transformation.)

We had incremental loading on the order of -50,000 kg/yr. in certain watersheds downstream of impoundments (FL6, and FL 9) that required us to use this offset. While this offset may seem large, incremental loads can reach above $10^6$ kg/yr (see Figure S7), so that it is not particularly influential. Note: the axis labels on Fig S7 should be "$* 10^6$", not "$+ 10^6$". This will be revised for the final manuscript.

Line 261. "ECs" need to be defined here. The acronym is only defined in the captions for Tables 2 and 3, but not in the main text.

We agree and will define "EC" at Line 189.

Line 620. "CI" needs to be defined as "credible interval" in the caption of Table 2.

We agree and will edit the text in the caption of Table 2.

.Supplementary material: Can the figure captions be checked to ensure that the captions contain all required information? E.g. It would be helpful for Figure S3 to define the dashed line in the caption, while dots and lines could be defined in the caption of Figure S4 (so that this is self-contained without relying on the caption of Figure 3).

We have checked all captions in the SI and will edit them so they are self-contained.

---

## Author Comment (AC2)

**Article:** Assessing inter-annual variability in nitrogen sourcing and retention through hybrid Bayesian watershed modeling

Response to Anonymous Referee #2: Responses in red

There are several areas of the manuscript that need improvements which require Major Revision.

(1) The abstract notes that the main contribution addresses the point that the "statistical calibration of loading models does not always yield plausible results"(lines 10-11) but it was difficult to see this aspect addressed in the manuscript as the broader contribution of the manuscript beyond the study area. However, later in the introduction it appears the main contribution is to improve the understanding of nitrogen export specifically for the two highly managed basins in North Carolina, USA (lines 45-46) using a smaller study with more dense monitoring than a previous study (Strickling and Obenour, 2018) over the same area (lines 40-53). This, in my opinion, is the weakest part of the paper and potentially makes it unsuitable for HESS. Strengthening is needed in the introduction to understand what broader research gap is being filled here, given the manuscript expands on an existing model of the study area.

Thank you for this critical feedback. We will edit the Introduction and Discussion to clarify the research contributions of this study. We now emphasize the limited capacity of previous hybrid modeling studies (including Strickling & Obenour 2018; Qian et al. 2005) to differentiate export rates among different source types (either they only included a small number of source types, or there was large overlap in their parameter estimates). Moreover, this study advances hybrid modeling (including Strickling & Obenour) by including interannual variability in both nutrient retention and export rates, which will be further emphasized.

While there is geographic overlap between Strickling and Obenour (2018), there is very little overlap in the data being used (only 2 of the monitoring sites are shared by both studies). We use a higher spatial resolution monitoring network than Strickling & Obenour (mean incremental watershed of 321 km$^2$ vs. 1535 km$^2$). At the same time, we hope the reviewer would agree that geographic overlap should not preclude a study from publication. We will revise the text to clarify this issue.

Later in lines 345-349, there are some statements that could indicate that these results could have potential for other studies related to nutrient loading due to agriculture. Perhaps normalizing your results by drainage area could make some of your results generalizable? In my reading, it seemed as though livestock played a smaller role because they occupy a smaller area of the basin.

Thank you for this suggestion. We will add normalized nutrient loading rates for livestock and compare them to the magnitude of agricultural TN export in each basin. For your reference, the table below shows the mean livestock export rate for the three basins:

| | kg/ha/yr | | | |
|---|---|---|---|---|
| | **Mean** | **ST dev** | **Min** | **Max** |
| **FL** | 0.25 | 0.08 | 0.12 | 0.42 |
| **HR** | 0.44 | 0.14 | 0.23 | 0.78 |
| **NHC** | 0.91 | 0.26 | 0.50 | 1.45 |

(2) There are numerous areas where the methods are not fully explained or choices are not justified. A reader would not be able to reproduce the study from the details provided solely in the manuscript.

We appreciate the reviewer's attention to methodological details, which can pose a challenge to any watershed modeling study of this spatial and temporal scope. At the same time, we think this manuscript provides a reasonable level of detail that compares well against other watershed modeling articles that we are familiar with. While not all modeling decisions can be extensively evaluated in a single manuscript, we have endeavored to be transparent. We hope that our responses to the detailed concerns (items a-p, below) will provide additional clarity.

(a) Line 66: The minimum data requirements for WRTDS seem incorrect. Please provide a citation here to support where you found the requirements to be a minimum of 5 years and 50 water quality samples. The original WRTDS paper (10.1111/j.1752-1688.2010.00482.x) states that one needs a minimum of 20 years and at least 200 samples.

We recognize this is an area of uncertainty with WRTDS, but hard requirements likely require more research, and that isn't the focus of this study. Temporal trend analysis in WRTDS requires a longer data series than just estimating loads (Chanat et al. 2013), and we are only doing the latter in this research. The WRTDS user-guide discusses minimum defaults and how to run the model with less data if needed, presumably so researchers can experiment with WRTDS in this way (Hirsch and De Cicco, 2015).

In addition, previous studies have analyzed yearly nutrient estimates for sites with a minimum of 5 years of data and nutrient trends with a minimum of 10 years (Fig. 2; Chanat et al., 2013). Of note, all, but one of our load monitoring sites has over 10 years of data.

We think our approach is reasonable, especially since we consider the estimated uncertainty of our yearly TN loading estimates based on the number of available observations (Section Fig S3). In the Methods, we will clarify that this effectively gives less weight to loading estimates derived from fewer samples.

While minimum data requirements for WRTDS are somewhat context-specific and may benefit from further research, the criteria used here are generally consistent with previous studies We will revise the text at line 67 to acknowledge that minimum data requirements are somewhat context specific.

(b) It is not clear what is the difference between incremental watershed and subwatershed throughout the manuscript (Section 2.3). What is each supposed to represent, in hydrologic terms? These need to be better defined.

We will add clarification between the two in the manuscript in Section 2.3. At a basic level, the total watershed is broken up into nested "incremental watersheds" at the locations of monitoring sites. This approach is well established in models like SPARROW, and we will add an appropriate reference (see below). Subwatersheds are subdivisions of the incremental watersheds, to more accurately account for nitrogen transport and retention (line 93). Both incremental watersheds and subwatersheds are shown in Figure '1, for reference.

Schwarz, G. E., Hoos, A. B., Alexander, R. B., & Smith, R. A. (2006). The SPARROW surface water-quality model: theory, application and user documentation. US geological survey techniques and methods report, book, 6(10), 248.

(c) Line 145: Explain and justify why the period of record was split into two and why that is a good choice.

Our text here was probably too vague.  LOADEST is a semi-parametric model that smooths loading estimates over time.  For substantial WWTP upgrade, we ran WRTDS separately for the period before and after the upgrade to avoid smoothing out the abrupt change in loading. We will clarify this in the text.  This was only necessary for a few stations (Table S2).

(d) Lines 149-151: How did you perform the preliminary analysis? I believe this belongs in the supplementary analysis.

Though we appreciate your interest in this topic, these WRTDS-specific questions are not the focus of this study and we think we have appropriately documented our procedure.  A more rigorous examination of how to handle data gaps in WRTDS would likely warrant its own manuscript.  In the WRTDS manual, it states that "A data gap of two years or less (regardless of the overall record length) is generally not a problem" (Hirsch and De Cicco, 2015), and we will note this consistency with manual guidance in our methods.

(e) Provide justification for equation (1) and why this calculation is needed as part of the workflow.

The result of this equation is used in Eq. 3. We will modify the text to help clarify this connection. In general, Eq. 1 characterizes the uncertainty in each incremental TN loading estimate, considering the uncertainties and correlations among upstream and downstream loadings that define the increment.  As noted in our response to your comment 'a', this equation essentially gives more weight to loading estimates that are based on a larger sample of observational data.

(f) Section 2.7 needs much more explanation. This seems to be the novel part of the work and what is different from Strickling and Obenour (2018). It was difficult to understand how the coefficients are determined and how negative loads are accounted for.

We think that this methods section is generally complete, but we will make additional edits for clarity.  In general, the coefficients are determined through Bayesian inference, and an extensive description of Bayesian methods is obviously not possible within manuscript page limits. We will also clarify that the offset was chosen to be greater than the largest negative loads (around -50,000 kg/yr).

(g) Line 203: I do not believe NHD+ is spelled out before its first use.

We agree and will define the National Hydrography Dataset Plus (NHD+) at this point in the manuscript.

(h) Eqn. 6: I do not think all of the variables are defined after the equation.

All variables in Eq. 6 are included in the text, but $r_{t,z}$ was defined incorrectly in Eq. 4 as $r_{i,z}$. This probably caused the confusion and will be fixed.

(i) Eqn 7a: I could not find where the calculation for tau(z) is described.

Mean residence time ($\tau_z$) was determined for each subwatershed and point source within GIS using the National Hydrography Dataset Plus (NHD+). This is explained at line 203.

(j) Line 212: I do not believe PIC is spelled out before its first use.

PIC was first defined as "precipitation impact coefficient" at line 190.

(k) Section 2.9: The selection of 4 different urban land-use splits is not well justified in the text. What is the hypothesis or scientific reasoning for the splits? Otherwise, it seems like the scenarios were made up with a trial and error to get the most attractive results.

We agree that we could have been clearer on why we used cutoffs of 1980 and 2000. The reason is because these cutoffs roughly represent urban areas built before and after changing NC environmental regulations related to erosion and sediment control (1980) and stormwater control measures (2000) that have come into effect over the past 50 years. We will edit the text to make this clearer in Section 2.9 and possibly in 4.1 if warranted.

(l) Lines 226-227: The "degree of overlap" was used to compare model fits but there is quantification given of this metric or objective reporting on why the particular scenario was selected.

In general, model "fit" was assessed primarily in terms of $R^2$. At the same time, we looked at the "degree of overlap" to determine if the model was able to distinguish between the different types of urbanization. If the coefficients for the urban categories were roughly similar (highly overlapping), there would be no strong evidence to support having variable urban loading rates. We generally consider a >95% probability as indicative of a high level of statistical confidence (roughly equivalent to $p < 0.05$ in frequentist statistics).  We acknowledge that there is some evidence for all scenarios, but there is the greatest

evidence for the pre/post 1980 split (e.g., Table 2). Along these lines, we will update some of the results text to ensure that the probabilities associated with degree of overlap are discussed clearly and consistently.

(m) Section 3.1 needs references to figures or tables to support the statements made here with figure and panel references.

All conclusions in Section 3.1 refer to Fig. 3 and S4, which are cited in the initial sentence. We will repeat this citation after other sentences for clarity in this section. We don't feel panel letters (or numbers) are efficient for Fig. 3 and S4 since we refer to each panel by watershed name in the text. We will clarify the captions for Fig. 3 and S4 to make them more consistent with the text of the manuscript.

(n) Section 3.2: A statistical test appear to be mentioned here but the test is not described nor is the null hypothesis so a reader cannot evaluate the validity of the test or the results.

Bayesian statistics bypasses the need for significance tests, which are arguably problematic for multiple reasons (see reference below, for example). At the same time, a probability of 95% or greater roughly corresponds to $p<0.05$ in a frequentist significance test (one-sided in this case). Since this manuscript was submitted to a special issue on Bayesian methods, we think the current language is generally appropriate. We will clarify that the difference/overlap in coefficients is based on samples from the posterior parameter distributions.

Cohen, J. (1994). The earth is round (p<. 05). American psychologist, 49(12), 997.

(o) Line 268: I do not believe EC is spelled out before its first use.

We agree and will define EC as "export coefficient" at its first use (line 189).

(p) Section 3.4: Few sentences offer supporting evidence for the statements made. Add references to figures or tables after each sentence to support these statements of fact. No proof appear

Reading Section 3.4, we are unsure which sentence you are referring to. There are 11 references within Section 3.4 already, and sentences without a reference are generally supported by a reference from the preceding sentence. We feel additional referencing could be excessive.

(3) The figures with multiple graphs need to have each panel labeled and referred to in the text. It was difficult to understand where to look for supporting evidence when only "Figure 2" is referenced but Figure 2 has 9 panels. This should be done for all figures with multiple panels.

We think the current row and column labels in gray are more efficient than individual panel labels in this case. To further assist the reader, we will add references to specific rows in this figure, where appropriate (e.g. line 109: (Fig. 2, top))

(4) It was difficult to follow the justification for the role (or lack thereof) that both soils (Section 3.4) and aging infrastructure (Section 4.1) play in this analysis. There seems to be

a lack of clear supporting evidence showing why or why not this is the case. There needs to be a more defined logical path in the text or these statements need to be removed.

Potential variations in TN export due to soil types was a question posed by local experts. It might not have as much appeal to an international audience and so we will cut the last paragraph of Section 3.4 that discusses watershed random effects and Triassic soils, as well as Fig S6 that showed major geologic regions in the region.

The discussion on aging infrastructure (Section 4.1) addresses why pre-1980 urban lands are likely to export increased levels on TN. We reference other studies that provide support for this hypothesis, and thus think it appropriate and important to leave this material in the manuscript.

(5) The data statement is no longer acceptable. It is now commonplace to have your data served on a publically available website. Even if HESS allows this outdated practice, major scientific organizations and publications - such as AGU - no longer allow statments that the data is available upon request. It is good practice -  - to serve the data using its own doi or as supplementary material.

All of the project datasets are available from documented public sources, as noted in our data statement. We note that other articles published in this Bayesian special issue of HESS include data statements similar to ours; we will defer to the editor.

Minor comments:

Line 217: Change to "distributions"

We will change this in the revised manuscript.

Line 380: Change to read "At the same time, loading attributable…"

We will change this in the revised manuscript.

**References**

Chanat, J. G., Moyer, D. L., Blomquist, J. D., Hyer, K. E., and Langland, M. J.: Application of a weighted regression model for reporting nutrient and sediment concentrations, fluxes, and trends in concentration and flux for the Chesapeake Bay Nontidal Water-Quality Monitoring Network, results through water year 2012 (No. 2015-5133), USGS, https://doi.org/10.3133/sir20155133, 2012.

Hirsch, R.M., and De Cicco, L.A.: User guide to Exploration and Graphics for RivEr Trends (EGRET) and dataRetrieval: R packages for hydrologic data (version 2.0): USGS Techniques and Methods book 4, chap. A10, https://doi.org/10.3133/tm4A10, 2015.

Qian, S. S., Reckhow, K. H., Zhai, J., & McMahon, G.: Nonlinear regression modeling of nutrient loads in streams: A Bayesian approach, Water Resour. Res., 41(7), https://doi.org/10.1029/2005WR003986, 2005.

Strickling, H.L. & Obenour, D.R.: Leveraging Spatial and Temporal Variability to Probabilistically Characterize Nutrient Sources and Export Rates in a Developing Watershed, Water Resour. Res, 54(7), 5143-5162, https://doi.org/10.1029/2017WR022220, 2018.

---

## Author Response (AR1)

Dear Dr. Glendall and Reviewers,

We are grateful for your positive, detailed, and constructive feedback. Thanks to your input, we believe that the manuscript has significantly improved in quality, especially as it relates to highlighting the novelty and wider relevance of the study. In general, all comments have been considered and we have made several changes to the manuscript as described in the individual responses below (with line numbering in responses referring to the revised manuscript with changes incorporated). Of note, we decided to change "inter-annual" to "interannual" in the manuscript title to be consistent.

On behalf of the authors, sincerely,

Jonathan Miller, PhD
North Carolina State University, Raleigh, NC.

**From the Editor:**

1. Can you please check a few terminology choices? 'Dischargers' may be best replaced with 'Point Sources' or 'Discharges'. Also 'hog' might be better replaced with 'pig'.

   Thank you for the suggestion. We have replaced Discharger with point source throughout the document. We also replaced "hog" with "swine" as we think this might be more appropriate for an international audience.

2. Definition of 'Hybrid watershed model' on line 30. I find that this concept is more clearly explained in Strickling and Obenour 2018 as a 'hybrid empirical/mechanistic' model. Could this perhaps be defined more clearly in the current manuscript?

   Thank you for the suggestion. We have modified the text at Lines 31-32.

3. Line 83 – HR should be introduced as abbreviation at first mention of Haw River

   Thanks, HR is defined at Line 64, and we now use this acronym consistently thereafter.

4. Please check consistent use of tenses, there appears to be a switch from past to present tense from section 3.2 onwards

   Thank you for noting this. We have made several edits for consistency. In general, we prefer to use present tense when presenting and discussion study results. At the same time, there are some cases when past tense is appropriate (e.g., describing historical changes in loading).

5. Line 346 – does 'TN produced by these animals' refer to the three major basins under study? If so, could this be clarified in the sentence please?
   We agree and have added "in our study area" at line 353.

**Anonymous Referee #1**

**General comments:**

This is a well-written paper, presenting an interesting model of nitrogen loading across river basins, accounting for temporal variability. In general, the methods appear to be appropriate, with assumptions and potential biases considered and appropriately accounted for, while the results are well interpreted and implications for policy are discussed.

I suggest below some specific comments, most of which are very minor in nature.

Thank you for the feedback on our manuscript; we address your specific comments below.

**Specific comments:**

Equation 1 (line 160). I am not sure I completely understand this formulation. It seems like there are only 2 upstream LMSs considered (k and l), while line 161 mentions "n". I do not have access to the original reference, but I wonder if some minor clarification would be helpful here.

"n" is a count variable that ranges from 1 to n. In our study, the largest value for n was 3 for site HR3. The "k and l" in the parentheses in this sentence confused this. We will clarify this in our revisions.

Section 2.9 (lines 216 to 232). Has a sensitivity analysis been carried out to investigate the effects of changing the informative priors? If not, I think this would be useful in understanding the robustness of the model. In any case, I think that some discussion of this is required.

In Strickling & Obenour (2018), such a sensitivity analysis was carried out on the data by running the hybrid watershed model with uninformative priors. This produced only small changes in the parameter estimates. In this study we are working with a larger observational dataset (25 load monitoring sites, compared to 21 sites in Strickling & Obenour), such that the influence of the priors is likely smaller. As such, we don't think this exercise would add new insights. We have noted this briefly in our revisions to the Results section at Lines 278-280.

Section 3.3 (lines 260 to 275). Can the CI endpoints be reproduced here? Currently, I feel that the point estimates without this context suggest greater certainty in these values than is the reality.

We agree that the point estimates might imply higher certainty than is warranted, but we do not want to duplicate all of the 95% CIs from Table 3, which might make the text cumbersome. Instead, we have added coefficients of variation (CV) in this section to convey the uncertainty in the point estimates, beginning at Line 273.

**Technical corrections:**

Line 46. A comma between "reservoirs" and "using" might be useful.

We have revised the text around Line 50, such that a comma is no longer needed.

Line 144. I think "plant" is unnecessary, being effectively a repetition here.

We agree and edited the text at line 150.

Line 172. Is $10^5$ definitely correct here? (It seems very large for an offset for a log transformation.)

We had incremental loadings on the order of -50,000 kg/yr in certain watersheds downstream of impoundments (FL6, and FL 9) that required us to use this offset. While this offset may seem large, incremental loads can reach above $10^6$ kg/yr (see Figure S7), so that it is not particularly influential. We added a brief note regarding these large negative loads at Line 183.

Line 261. "ECs" need to be defined here. The acronym is only defined in the captions for Tables 2 and 3, but not in the main text.

We agree and have defined "EC" at Line 198.

Line 620. "CI" needs to be defined as "credible interval" in the caption of Table 2.

We agree and have edited the text in the caption of Table 2.

.Supplementary material: Can the figure captions be checked to ensure that the captions contain all required information? E.g. It would be helpful for Figure S3 to define the dashed line in the caption, while dots and lines could be defined in the caption of Figure S4 (so that this is self-contained without relying on the caption of Figure 3).

We have checked all captions in the SI and have edited them so they are self-contained.

**Anonymous Referee #2**

There are several areas of the manuscript that need improvements which require Major Revision.

(1) The abstract notes that the main contribution addresses the point that the "statistical calibration of loading models does not always yield plausible results"(lines 10-11) but it was difficult to see this aspect addressed in the manuscript as the broader contribution of the manuscript beyond the study area. However, later in the introduction it appears the main contribution is to improve the understanding of nitrogen export specifically for the two highly managed basins in North Carolina, USA (lines 45-46) using a smaller study with more dense monitoring than a previous study (Strickling and Obenour, 2018) over the same area (lines 40-53). This, in my opinion, is the weakest part of the paper and potentially makes it unsuitable for HESS. Strengthening is needed in the introduction to understand what broader research gap is being filled here, given the manuscript expands on an existing model of the study area.

Thank you for this critical feedback. We have edited the Introduction and Discussion to clarify the research contributions of this study. We now emphasize the limited capacity of previous

Bayesian hybrid modeling studies (e.g., Qian et al. 2005; Wellen et al., 2012; Strickling & Obenour 2018) to differentiate export rates among different source types (either they only included a small number of source types, or there was large overlap in their parameter estimates). This is now highlighted at Lines 10-13, 45-49. This theme is also addressed in the discussion, including at Lines 415-420.

Moreover, this study advances hybrid modeling by including interannual variability in both nutrient retention and export rates, which is now further emphasized at Line 53. This theme is returned to in the Discussion (Section 4.2).

While there is geographic overlap between Strickling and Obenour (2018), there is very little overlap in the data being used (only 2 of the monitoring sites are shared by both studies). We use a higher spatial resolution monitoring network than Strickling & Obenour (mean incremental watershed of 321 $km^2$ vs. 1535 $km^2$). This is now emphasized at Lines 55-56 and 418-421. At the same time, we hope the reviewer would agree that geographic overlap should not preclude a study from publication.

Later in lines 345-349, there are some statements that could indicate that these results could have potential for other studies related to nutrient loading due to agriculture. Perhaps normalizing your results by drainage area could make some of your results generalizable? In my reading, it seemed as though livestock played a smaller role because they occupy a smaller area of the basin.

Thank you for this suggestion. We normalized nutrient loading rates for livestock and compare them to the magnitude of agricultural TN export in each basin. Our expanded discussion can now be found at lines 354-358.

(2) There are numerous areas where the methods are not fully explained or choices are not justified. A reader would not be able to reproduce the study from the details provided solely in the manuscript.

We appreciate the reviewer's attention to methodological details, which can pose a challenge to any watershed modeling study of this spatial and temporal scope. At the same time, we think this manuscript provides a reasonable level of detail that compares well against other watershed modeling articles that we are familiar with. While not all modeling decisions can be extensively evaluated in a single manuscript, we have endeavored to be transparent. We hope that our responses to the detailed concerns (items a-p, below) will provide additional clarity.

(a) Line 66: The minimum data requirements for WRTDS seem incorrect. Please provide a citation here to support where you found the requirements to be a minimum of 5 years and 50 water quality samples. The original WRTDS paper (10.1111/j.1752-1688.2010.00482.x) states that one needs a minimum of 20 years and at least 200 samples.

We recognize this is an area of uncertainty with WRTDS, but hard requirements likely require more research, and that isn't the focus of this study. Temporal trend analysis in WRTDS requires a longer data series than just estimating loads (Chanat et al. 2013), and we are only doing the latter in this research. The WRTDS user-guide discusses minimum defaults and how to run the

model with less data if needed, presumably so researchers can experiment with WRTDS in this way (Hirsch and De Cicco, 2015).

In addition, previous studies have analyzed yearly nutrient estimates for sites with a minimum of 5 years of data and nutrient trends with a minimum of 10 years (Fig. 2; Chanat et al., 2013). Of note, all, but one of our load monitoring sites has over 10 years of data.

We think our approach is reasonable, especially since we consider the estimated uncertainty of our yearly TN loading estimates based on the number of available observations (Section Fig S3). In the Methods, we now clarify that this effectively gives more weight to loading estimates derived from more nutrient samples (Line 162).

While minimum data requirements for WRTDS are somewhat context-specific and may benefit from further research, the criteria used here are generally consistent with previous studies We revised Line 72 to acknowledge that minimum data requirements are somewhat context specific.

(b) It is not clear what is the difference between incremental watershed and subwatershed throughout the manuscript (Section 2.3). What is each supposed to represent, in hydrologic terms? These need to be better defined.

We have made edits in Section 2.3 to help clarify. At a basic level, the total watershed is broken up into nested "incremental watersheds" at the locations of monitoring sites. This approach is well established in models like SPARROW, and we will add an appropriate reference (see below). Subwatersheds are subdivisions of the incremental watersheds, to more accurately account for nitrogen transport and retention (Line 93). Both incremental watersheds and subwatersheds are shown in Figure 1, for reference.

Schwarz, G. E., Hoos, A. B., Alexander, R. B., & Smith, R. A. (2006). The SPARROW surface water-quality model: theory, application and user documentation. US geological survey techniques and methods report, book, 6(10), 248.

(c) Line 145: Explain and justify why the period of record was split into two and why that is a good choice.

Our text here was probably too vague.  LOADEST is a semi-parametric model that smooths loading estimates over time.  For substantial WWTP upgrade, we ran WRTDS separately before and after the upgrade date to avoid smoothing out these transitions. We clarify this in the text at Line 150.  This was only necessary for a few stations (Table S2).

(d) Lines 149-151: How did you perform the preliminary analysis? I believe this belongs in the supplementary analysis.

Though we appreciate your interest in this topic, these WRTDS-specific questions are not the focus of this study and we think we have appropriately documented our procedure.  A more rigorous examination of how to handle data gaps in WRTDS would likely warrant its own manuscript.  In the WRTDS manual, it states that "A data gap of two years or less (regardless of

the overall record length) is generally not a problem" (Hirsch and De Cicco, 2015), and we note this consistency with manual guidance in Line 156.

(e) Provide justification for equation (1) and why this calculation is needed as part of the workflow.

The result of this equation is used in Eq. 3. We modified the text to help clarify this connection at Lines 161-165. In general, Eq. 1 characterizes the uncertainty in each incremental TN loading estimate, considering the uncertainties and correlations among upstream and downstream loadings that define the increment. As noted in our response to your comment 'a', this equation essentially gives more weight to loading estimates that are based on a larger sample of observational data.

(f) Section 2.7 needs much more explanation. This seems to be the novel part of the work and what is different from Strickling and Obenour (2018). It was difficult to understand how the coefficients are determined and how negative loads are accounted for.

We think that this methods section is generally complete, but we made additional edits for clarity. In general, the coefficients are determined through Bayesian inference, and an extensive description of Bayesian methods is obviously not possible within manuscript page limits. We also clarify that the offset was chosen to be greater than the largest negative loads (around -50,000 kg/yr) in Lines 182-184.

(g) Line 203: I do not believe NHD+ is spelled out before its first use.

We agree and defined the National Hydrography Dataset Plus (NHD+) in Line 212.

(h) Eqn. 6: I do not think all of the variables are defined after the equation.

All variables in Eq. 6 are included in the text, but $r_{t,z}$ was defined incorrectly in Eq. 4 as $r_{i,z}$. This probably caused the confusion and is now fixed on Line 189.

(i) Eqn 7a: I could not find where the calculation for tau(z) is described.

Mean residence time ($\tau_z$) was determined for each subwatershed and point source within GIS using the National Hydrography Dataset Plus (NHD+). This is explained at Line 210-213.

(j) Line 212: I do not believe PIC is spelled out before its first use.

PIC was first defined as "precipitation impact coefficient" at Line 199.

(k) Section 2.9: The selection of 4 different urban land-use splits is not well justified in the text. What is the hypothesis or scientific reasoning for the splits? Otherwise, it seems like the scenarios were made up with a trial and error to get the most attractive results.

We agree that we could have been clearer on why we used cutoffs of 1980 and 2000. The reason is because these cutoffs roughly represent urban areas built before and after changing NC environmental regulations related to erosion and sediment control (1980) and stormwater control measures (2000) that have come into effect over the past 50 years. We edited the text to make this clearer in Section 2.9 (Lines 234-236).

(l) Lines 226-227: The "degree of overlap" was used to compare model fits but there is quantification given of this metric or objective reporting on why the particular scenario was selected.

In general, model "fit" was assessed primarily in terms of $R^2$. At the same time, we looked at the "degree of overlap" to determine if the model was able to distinguish between the different types of urbanization. If the coefficients for the urban categories were roughly similar (highly overlapping), there would be no strong evidence to support having variable urban loading rates. We generally consider a >95% probability as indicative of a high level of statistical confidence (roughly equivalent to $p<0.05$ in frequentist statistics). We acknowledge that there is some evidence for all scenarios, but there is the greatest evidence for the pre/post 1980 split (e.g., Table 2). Along these lines, we updated section 3.2 (Lines 266-267) to ensure that the probabilities associated with degree of overlap are discussed clearly and consistently.

(m) Section 3.1 needs references to figures or tables to support the statements made here with figure and panel references.

All conclusions in Section 3.1 refer to Fig. 3 and S4, which are cited in the initial sentence. We added this citation after other sentences for clarity in this section. We don't feel panel letters (or numbers) are efficient for Fig. 3 and S4 since we refer to each panel by watershed name in the text. We also clarified the captions for Fig. 3 and S4 to make them more consistent with the text of the manuscript.

(n) Section 3.2: A statistical test appear to be mentioned here but the test is not described nor is the null hypothesis so a reader cannot evaluate the validity of the test or the results.

Bayesian statistics bypasses the need for significance tests, which are arguably problematic for multiple reasons (see reference below, for example). At the same time, a probability of 95% or greater roughly corresponds to $p<0.05$ in a frequentist significance test (one-sided in this case). Since this manuscript was submitted to a special issue on Bayesian methods, we think the current language is generally appropriate. We clarified that the difference/overlap in coefficients is based on samples from the posterior parameter distributions in Lines 263-64 and 267.

Cohen, J. (1994). The earth is round (p<. 05). American psychologist, 49(12), 997.

(o) Line 268: I do not believe EC is spelled out before its first use.

We agree and defined EC as "export coefficient" at its first use (Line 198).

(p) Section 3.4: Few sentences offer supporting evidence for the statements made. Add references to figures or tables after each sentence to support these statements of fact. No proof appear

Reading Section 3.4, we are unsure which sentence you are referring to. There are 11 references within Section 3.4 already, and sentences without a reference are generally supported by a reference from the preceding sentence. We feel additional referencing could be excessive.

(3) The figures with multiple graphs need to have each panel labeled and referred to in the text. It was difficult to understand where to look for supporting evidence when only "Figure 2" is referenced but Figure 2 has 9 panels. This should be done for all figures with multiple panels.

We think the current row and column labels in gray are more efficient than individual panel labels in this case. To further assist the reader, we added references to specific rows in this figure at Lines 115, 124, and 138.

(4) It was difficult to follow the justification for the role (or lack thereof) that both soils (Section 3.4) and aging infrastructure (Section 4.1) play in this analysis. There seems to be a lack of clear supporting evidence showing why or why not this is the case. There needs to be a more defined logical path in the text or these statements need to be removed.

Potential variations in TN export due to soil types was a question posed by local experts. It might not have as much appeal to an international audience and so we cut the last paragraph of Section 3.4 that discusses watershed random effects and Triassic soils, as well as Fig S6 that showed major geologic regions in the region.

The discussion on aging infrastructure (Section 4.1) addresses why pre-1980 urban lands are likely to export increased levels on TN. We reference other studies that provide support for this hypothesis, and thus think it appropriate and important to leave this material in the manuscript.

(5) The data statement is no longer acceptable. It is now commonplace to have your data served on a publically available website. Even if HESS allows this outdated practice, major scientific organizations and publications - such as AGU - no longer allow statments that the data is available upon request. It is good practice - - to serve the data using its own doi or as supplementary material.

We have added our RStan code to the SI and will make our dataset available on a public repository. We have changed our data statement to: "Hydrometeorological and water quality data can be obtained from the public sources described in the methods (e.g., USGS, Water Quality Portal). RStan code is included in the SI (Text S1). A complete input dataset for the code is located here: "

**Minor comments:**

Line 217: Change to "distributions"

We changed this in the revised manuscript at Line 226.

Line 380: Change to read "At the same time, loading attributable…"

We changed this in the revised manuscript at Line 386.

**References**

Chanat, J. G., Moyer, D. L., Blomquist, J. D., Hyer, K. E., and Langland, M. J.: Application of a weighted regression model for reporting nutrient and sediment concentrations, fluxes, and trends in concentration and flux for the Chesapeake Bay Nontidal Water-Quality Monitoring Network, results through water year 2012 (No. 2015-5133), USGS, https://doi.org/10.3133/sir20155133, 2012.

Hirsch, R.M., and De Cicco, L.A.: User guide to Exploration and Graphics for RivEr Trends (EGRET) and dataRetrieval: R packages for hydrologic data (version 2.0): USGS Techniques and Methods book 4, chap. A10, https://doi.org/10.3133/tm4A10, 2015.

Qian, S. S., Reckhow, K. H., Zhai, J., & McMahon, G.: Nonlinear regression modeling of nutrient loads in streams: A Bayesian approach, Water Resour. Res., 41(7), https://doi.org/10.1029/2005WR003986, 2005.

Strickling, H.L. & Obenour, D.R.: Leveraging Spatial and Temporal Variability to Probabilistically Characterize Nutrient Sources and Export Rates in a Developing Watershed, Water Resour. Res, 54(7), 5143-5162, https://doi.org/10.1029/2017WR022220, 2018.